# Seismic Composite Metamaterial: A Review

**Al-Shami Qahtan** [1,2,3], **Jiankun Huang** [1,2,*], **Mugahed Amran** [3,4,*], **Diyar N. Qader** [5], **Roman Fediuk** [6,7,*] and **Al-Dhabir Wael** [8]

1 Key Laboratory of State Forestry Administration on Soil and Water Conservation, School of Soil and Water Conservation, Beijing Forestry University, Beijing 100083, China
2 School of Soil and Water Conservation, Beijing Forestry University, Beijing 100083, China
3 Department of Civil Engineering, Faculty of Engineering and IT, Amran University, Amran 9677, Yemen
4 Department of Civil Engineering, College of Engineering, Prince Sattam Bin Abdulaziz University, Alkharj 11942, Saudi Arabia
5 Department of Civil Engineering, Cihan University-Erbil, Erbil 44001, Kurdistan Region, Iraq
6 Polytechnic Institute, Far Eastern Federal University, Vladivostok 690922, Russia
7 Department of Ship Energy and Automation, Peter the Great St. Petersburg Polytechnic University, St. Petersburg 195251, Russia
8 Civil Engineering Department, Faculty of Engineering, Sakarya University of Applied Sciences, Kemalpaşa 54050, Sakarya, Turkey
* Correspondence: jiankunhuang@bjfu.edu.cn (J.H.); m.amran@psau.edu.sa (M.A.); fedyuk.rs@dvfu.ru (R.F.)

**Abstract:** The modern construction revolution throughout the past two decades has brought the need for ground vibration mitigation, and this has been one of the major study areas. These studies were mainly focused on the effect of forestation on vibration reduction as the available natural metamaterial. Physical methods such as the finite element method and the boundary conditions of 2D and 3D applications in ground vibration reduction have been developed. Many researchers, scientists, and organizations in this field have emphasized the importance of these methods theoretically and numerically. This paper presents the historical context of resonant metamaterials (MMs), the current progress of periodic 2D and 3D structures, and the possible future outcomes from the seismic metamaterials (SMs), and it relates them with their elastic counterparts to the natural metamaterial (NMs). The idea of bandgaps (FBGs) in the frequency range of interest is reviewed and discussed in some detail. Moreover, the attenuation associated with ground vibrations, noise, seismology, and the like is explained by managing the peculiar mechanisms of ground vibrations. However, a comprehensive computational review focuses on shielding MMs for ground vibration mitigation in urban areas. This phenomenon led to unique features for various techniques to control the bandgap width for various construction applications. Ecological solutions involve the creation of an economic, environmentally based seismic shield for both the Bragg scattering and the local resonance bandgaps. Reportedly, additive studies based on numerical simulation and experiments have improved the functionality of the 2D and 3D periodic structures. It was found that the mechanical properties differ (i.e., stiffness, Poisson's ratio, and bulk density) and that the geometrical parameters (i.e., lattice, model dimensions, distance from vibration sources, and number of periodic structures) exhibited strong effects on the width and location of the derived FBGs. The geometrical properties of the used unit cell have a strong effect on the attenuation mechanism. Although deep analysis was created in much of the previous research, it was revealed, based on that research, that the attenuation mechanism is still unclear. However, this review article presents a detailed exposition of the recent research progress of the seismic metamaterials, including 2D, 3D, and the main mechanisms of the theoretical backgrounds of energy attenuation. It also summarizes the effects of the factors on the width and location of the bandgaps at a low frequency. In addition, the natural metamaterials and the study of the urban environment are surveyed. The major findings of this review involve the effectiveness of NMs for different functionalities in ground vibration attenuation, which leads to diverse purposes and applications and proposes a roadmap for developing natural materials for clean and quiet environments.

**Keywords:** composite; seismic; metamaterial; ground vibration; periodic structure; low frequency; forest tree

## 1. Introduction

People live quietly in a vibrating world with a lot of man-made noise pollution and ground vibrations, including those caused by industries, mining, and traffic. Ground vibration exposure causes more sleeplessness and cardiac problems, with the vibrations being particularly important. There has been a gradual improvement in residents' requirements for the quality of the urban environment [1]. Seismic metamaterials (SMs) are a special type of engineered metamaterials that are characterized by an elastic resonance response, and they are specifically designed to impede unwanted ground vibrations; they not only impede the vibrations resulting from natural disasters and earthquakes, but also help in the reflection, absorption, refraction, damping, redirection, and mitigation of the vibrations resulting from the human activities due to traffic movement, transportation, factories, mining, and various construction activities [2]. Seismic wavelengths range from decimeters to hectometers. Seismic waves are produced by polarizing the movement of tectonic plates in the form of particle waves decomposing in the ground, and some of them reach the surface of the earth in the form of surface waves, namely Rayleigh waves and Love waves, which contain energy that causes destructive damage to buildings and facilities and destabilizes them and threatens human life [3]. There is a lack of consideration of ground vibration environmental pollution when planning cities in most countries of the world, which makes the houses near the main traffic lines vulnerable to this type of pollution. The residents in those buildings become the victims; so, living in those buildings is demanding, and it is boring and uncomfortable in all of the houses in the community. Effectively reducing the impact of traffic noise and vibration on the surrounding environment is one of the major issues of the researchers and engineers concerned with the engineering of the national traffic environment [4–7]. Therefore, necessary measures should be taken to reduce vibration and noise, which have become the most serious pollution sources affecting people's lives [8,9]. According to the latest statistics, more than 60% of the complaints about environmental pollution are about noise pollution, and more than half of them are about traffic noise [10,11]. Noise and vibration pollution are a cause of people's physical and psychological discomfort [12,13]. All kinds of ground vibrations caused by earthquakes, elastic waves, acoustic sources, human activities, and mining and all kind of traffic trains, automobiles, and urban rail transit noise emit vibration frequencies ranging from tens of hertz to thousands of hertz; these are broadband pollution sources [14]. The existing research has shown that low-frequency noise can cause negative effects as close to the start as 40 dB. Some low-frequency vibrations, such as those between 10 and 100 Hz, which are near to the natural frequency, can be annoying. Anger has been linked to subjective perceptions of fatigue, drowsiness, and loss of attention [15]. In addition, some high-frequency noises, such as those at 2.5–3.5 kHz, cause direct damage to the auditory organs. High-frequency hearing loss cannot be cured but can be prevented. Therefore, it is everyone's duty to identify preventive measures to protect against and mitigate noise exposure [16,17]. The main reasons for the increasing traffic noise pollution are the sharp increase in the number of cars, the development of transportation engineering, and the road projects through the city [18]. The second is the congestion caused by the interruption of the transportation path and the congestion of vehicles that may result in vibration and noise [19]. Although the vibrations and noise from road traffic will not cause direct damage to buildings, they may cause local tremors in the internal structure of the buildings and even create secondary structural noise in the buildings [20]. In the pursuit of fulfilling the desires for urban planning, addressing various convenient services in the city, such as transportation, housing, and other services, helping to create a beautiful view of the city and improving it, and contributing to providing a base for human activity through the

conservation and exploitation of land and the proper use of land, it is particularly common in practical engineering applications to set up sound barriers, vibration-damping piles, and vibration-damping trenches in the transmission paths of vibration and noise. The vibration and noise reduction measures for the protected object include three aspects. First, in the design, the building can have a wide foundation; vibration isolation pads [21,22], vibration isolation supports [23,24], and other passive vibration isolation systems can be added [25–27]; a spring damping system on the vibration body [28] and the supporting structure to change the vibration characteristics of the entire structure can be installed [29], thereby reducing the impact of vibration on the buildings and precision instruments or cultural relics [29]. Second, for existing buildings, sound insulation or sound-absorbing materials such as lightweight aggregate concrete [30,31], can be attached to the surface of the building [30], or sound-insulating windows can be added to the interior of the building [32]. Common sound insulation window materials include wood structures [33], steel structures [34], and aluminum alloy structures [35]. The third is to take certain protective measures for the protected object to isolate the noise and vibration [36]. The existing studies have focused on vibration reduction measures, such as sound barriers [37], green belts [38–47], or sound-absorbing ceilings on both sides of the road [48]. Most of the sound barrier structures are often porous structures [49], such as perforated plates [50], foam glasses [51], etc. These sound-absorbing boards are widely used due to their simple production and low cost. Acoustic materials have a narrow sound absorption frequency band, and it is impossible to reduce noise with a simple sound barrier [52]. In recent years, there has been an in-depth study of periodic structures. Irwin et al. [53] proposed the first idea of using single-row or multi-row, thin-walled circular holes as wave barriers. Subsequently, Yang X. et al. [54] conducted some domestic in-depth research on discontinuous barriers; their research had three aspects, theoretical research, numerical simulation, and experimental verification, which proved the good vibration isolation performance of the barriers. The application of some artificially designed periodic foundations, underground piles, and wave barriers in civil engineering vibration reduction and earthquake resistance has also been studied in depth [55]. Hall et al. (2003) established a comparison link between the velocity of ground vibrations and the noise level on the facade. Even if there are no traffic jams or traffic violations, any passing vehicle would cause a form of ground stress [54]. Zhang et al. [56] established simulations and analyses to form a link between the velocity of ground vibrations and the noise level on the facade in the finite element method in order to induce train vibrations. Since then, research has been conducted to determine the human response to railway-induced vibrations [57], which are generally overlooked in comparison to ground vibrations. Ogren et al. [58] compared the irritation produced by the vibrations of railways and the noise. It is crucial to understand how people who have been exposed to vibration feel about it and how much discomfort it causes in their homes in order to consider the reduction measures for the ground vibrations. In general, these vibration activities can be attenuated by reducing the incoming vibrations.

The use of different systems of seismic metamaterials to suppress or redirect waves has been the focus of many researchers and academics recently. Some of these newly developed systems are simple lenses and mirrors, which are used to redirect and focus electromagnetic radiation at optical wavelengths, and they represent continuous attempts to influence wave propagation, while the application of seismic lensing by altering the ground's refractive index has only recently been reviewed [59]. Many investigations have employed and developed waveguides in which the dispersion relation indicates bandgaps, also known as stop bands or filter bands; these are the ranges of frequencies in which waves cannot travel through the material. Meta-surfaces [60,61], acoustics [62,63], elastics [64,65], and waveguides [66] have all been developed. In the ground vibrations, different models of periodic structures are solved using Green's equation. Low-frequency vibrations are the most difficult to reduce since the earth does not dampen them much; many variables are still being assessed. Krylov et al. [67] evaluated sleepers as line barriers when arranged to interact with the ground vibrations from railway sources. Hunt et al. [68] followed up on this

research. Svinkin et al. [69] proposed that ground vibrations were affected by the geotechnical properties of soil. These models were theoretically investigated based on attributes of ground vibrations as well as ground parameters. Lastly, Colace et al. [70] investigated the proposition that a substantial rise in vibration levels was due to an increase in the vehicles' unsprung mass. The vibrations caused by human activity not only impair sensible structures, they also have a negative impact on individuals. As human activities increase more and more throughout cities, people are more worried about quality and comfort. The increase in complaints about noise and vibrations has led to more interest in developing different systems. Base isolation mitigation systems can be used at the foundations to protect the whole building from the harm of ground vibration. Di Matteo et al. [71] investigated the combination of three passive control systems to evaluate the plane wave response of base isolation systems. They found that the mitigation techniques, when used together, are inefficient. Jiang et al. [72] investigated train vibration mitigation models and applied them on a broad scale using in-filled or open trenches and using special materials that form a vibration mitigation system when combined with the ground. According to a numerical simulation, the use of wave barriers made of seismic metamaterials that have a stiffness higher than the soil-medium stiffness can be more effective, especially when the differences in stiffness between them are adequately higher [73]. Huang et al. [74] investigated the high thickness and long distance from one neighboring wall to another neighboring wall; the result was a higher supplement loss, especially when those procedures are close to 25% of the wavelength of the wave. Coulier and Hunt [75] validated this in a laboratory that used gelatin in place of dirt to minimize the wavelengths in the test scale. After full-scale experiments, the subsurface barriers' success in actual vibration was discovered. Huang and Zhifei Shi [55] conducted numerical research on pile barrier analysis and design for block vibrations, particularly in the low-frequency range. Dijckmas et al. [76] evaluated the heavy mass efficiency when located above the earth surface in an array continuously around the track. This method of wall barriers is valuable for the reduction in unwanted ground vibration. Subsequently, they looked at how a sheet pile wall was successfully used to mitigate ground vibrations. They came to the conclusion that porous walls can be employed as vibration barriers, with the stiffness of these walls and the depth of the soil determining the efficacy of the reduction mechanism. In other work, they discovered that heavy biomasses/masses, when placed above the earth's surface, reduce incident surface waves at resonance frequencies [77]. Persson et al. [78] looked at the impacts of water infiltrations on the open trenches and found that when there is a considerable volume of infiltrated water present, the trench's efficacy reduces because the water permits the primary waves to transmit. When the water tables are adequate, the trenches can be adequate; the trench's efficacy is reduced when the depth of the trench is reduced from 16 m deep to 12 m. As a result, the vibration levels will be reduced from 65% to 21%. Hoorickx et al. [64] investigated the behavior of double and single jet-grouted wave barriers made of the same materials and volumes; they found that the dual-wall baffles behaved better at short spacing along the barriers. Huang et al. [79] suggested a wave barrier of multi-layered periodic structures containing two layers of diversely changed components; they found that the attenuation mechanism was greatly influenced by the depth and number of rows of the periodic barriers. Bordón et al. [80] explored the optimization of the forming, inclination, location, thickness, and tilt of single and dual walls; they found that at a wall depth of less than 110% of the wavelength, no significant improvements were observed due to barrier topology. On the other hand, by inclining and relocating a wall, there was more efficiency in comparison to the normal case. Yarmohammadi et al. [81] concluded that the mitigation capacities of open trenches are higher than those of in-filled trenches, and in order to obtain more than a 20% increase in the mitigation capacity, the double trench barriers should be used instead of the single ones. However, a three-tiered trench barrier has no significant impact on level mitigation. The ground vibration mitigation through an urban environment has been one of the major study areas in the modern construction revolution throughout the past two decades. These studies were mainly focused on the

effect of forestations as an available natural metamaterial on vibration reduction. Different methods (such as boundary conditions, finite element analysis, the sound cone method, and the strain energy density method) of 2D and 3D applications in noise and ground vibration reduction have been developed. Many researchers, scientists, and organizations in this field have emphasized the importance of this method.

This paper focuses on shielding SMs for ground vibration mitigation and noise control, as well as the unique features of the various techniques in engineering bandgaps. In the context of this review, the role of periodic structures for processing seismic waves at the macro-scale is clarified to protect the engineering infrastructure from unwanted ground vibrations. Moreover, the research model has shifted from the structural materials used in SMs to those of NMs, rearranging the periodic structures in the soil in a specific geometric way, The capability to generate Bragg scattering and local resonance bandgaps is part of the NM's open range of applications toward urban environments. Within the framework of the great interest in the research activities that have recently spread out from resonant MMs, it was found that urban areas have an effective role in the mitigation of ground vibrations. This paper reviews the effectiveness of NMs in shaping and impeding the flow of seismic waves and vibrations in the low-frequency range and summarizes the influencing factors from previous studies in order to contribute to the suggesting of models. Being more effective involves meeting the current technological challenges and overcoming them in the future by investigating the ability of NMs to attenuate propagation in order to find the effects of key factors for the future prospects and to experimentally and numerically develop models that provide more ideas for understanding the further features of the forest in order to bring environment solutions for the several issues produced due to human activities and natural disasters.

## 2. Overview Periodic Structure Theory

The term "periodic structure" is used to describe an arrangement of units with similar properties in space that is endlessly repeated in accordance with a predetermined pattern. One-dimensional, two-dimensional, and three-dimensional are the three types of periodic structures that are mainly classified based on the spatial arrangements of typical units. According to this argument, as shown in Figure 1, one-dimensional periodic structures typically take the form of layers or rods made of two or more materials, whereas two-dimensional periodic structures often consist of lattice nodes, periodic arrays, or rows of piles; furthermore, periodic lattice structures, which are also known as 3D periodic structures, often have the shape of a body or a simple, centered cubic lattice.

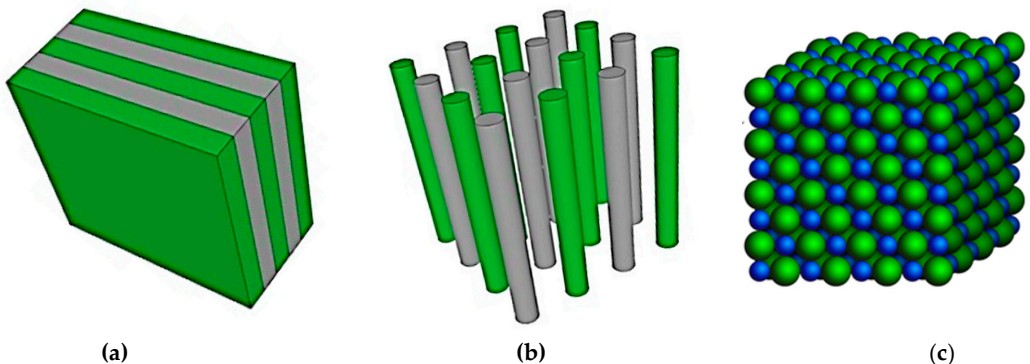

(a)            (b)            (c)

**Figure 1.** Diagrams of periodic structures in different dimensions: (**a**) one dimension; (**b**) two dimensions; (**c**) three dimensions.

The first idea of periodic structures appeared in engineering recently, in the last decade of the last century; it was within the principles of MMs and phononic crystal, which were then were commonly used in the fields of solid physics, especially in the microscopic fields. After that, many techniques were developed based on the concepts of MMs, which aim to obstruct EWs and absorb the wave energy in some bands of frequencies (bandgaps).

As a result, interests have turned toward the discovery of MMs with resonant properties for use as a seismic shield; this has particular relevance because it has the potential to protect entire sites and groups of buildings from ground vibrations and well-designed source interventions above the ground surface [82]. Buried structures can provide some solutions for the same purpose, but there are some challenges regarding their performance because their efficiency is largely dependent on their depth, especially at low frequencies. Barbosa et al. [83] showed the success of using stiff barriers to protect in-filled trenches, mainly by a wave-directing effect caused by the interaction between the propagating surface waves in the ground and the bending of the barriers propagating the waves.

The qualities of the MMs and their initial material, shape, dimensions, and properties are the focus of previous studies [80,81]. On the other hand, all of those structures are set at a distance from the sources of vibration and are constructed in a continuous manner to protect specific areas [84]. In recent years, phononic crystals have been investigated for their potential as periodic structures to reduce vibrations [85]. These structures were first investigated as acoustic barriers, and they are a relatively new invention [84–86]. This concept's application in mitigating and controlling vibrations is considerably more concerning than the topics that are still in the study's interests. Castan-heira Pinto et al. [87] evaluated the impact of continuous barriers of periodic structures buried parallel along the railway and discovered a very complex response pattern. They also confirmed that the efficiency of the presence of periodic structures is affected by their distribution in the form of groups. By extending the periodic structure from the nanoscale to the macroscopic scale and replacing the periodically arranged atoms with elastic scatterers, the effect of the periodic structure on the elastic wave has a similar effect to that of the atomic periodic potential field on electrons. Because of this similarity, it is appropriate to use lattice theory in solid-state physics to directly describe the frequency of the scattering structure [88]. Periodic structures have numerous practical applications in a variety of fields, including elastic response, electromagnetism, waveguiding, and sound sensors, among others [89]. Periodic structures, such as an array of periodic barriers and in-filled trenches, have the ability to enhance elastic wave absorption, thereby enhancing the performance of applications such as seismic wave attenuation, efficiency, and absorbing interfaces [90]. In addition, research into periodic structures is an ongoing undertaking, within which innovative methodologies and analyses might be sought for related applications. This section discusses the analysis, diagnosis, and improvement of 1D, 2D, and 3D periodic structures for applications involving ground vibration and the low frequency of seismic waves, as well as SM technology. In particular, a novel technique to array the layout of NMs will be introduced to prevent the emergence of seismic wave propagation and to reduce low-frequency ground vibration. In fact, this paper does not cover the investigation of the concept of periodic structures in the crystal, but it is valuable in controlling the elastic wave in the future.

### 2.1. Lattice Space Dimension

In terms of wave propagation modeling, the finite element method and the boundary conditions are the most commonly used techniques, with an emphasis on wave manipulations in 2D and 3D space for the different types of guided waves associated with particular applications. The production of an FBG, as well as the control of its breadth and localization within the band structure, has long been of interest to the scientists who study periodic structures and SMs. The purpose of this section is to examine the attenuation process underlying their origin, focusing on 2D and 3D designs to mimic the subwavelength FBG manifestation. Numerical models are preferred because of the complexity of wave propagation, the high cost of field tests or even full-scale experiments, and their superior computational efficiency in forecasting the ground vibrations caused by vibratory sources. FBG production has therefore emerged as a result of its use in applications such as vibration mitigation, seismic shielding, multidirectional wave cancellation, waveguiding, and sound sensors. The understanding of FBGs and their application has resulted in a variety of SM

designs, particularly for 2D and 3D lattices. Specifically, the management of guided waves, such as Love and Rayleigh waves, has prompted the study of periodic structures.

### 2.1.1. Two-Dimensional Metamaterials

In recent years, the interest in FBG engineering has increased rapidly because of its advantages in terms of high performance and the implementation of elastic wave propagation. The early research on the elastic FBG was conducted on 2D periodic structures, predominantly of cylindrical shape, in a medium housing the dispersion of bulk waves [91,92]. The purpose of this part is to illustrate a periodic mismatch between the generated FBGs and the 2D structures. Numerous studies have focused on two-dimensional arrangements to duplicate the subwavelength FBG. Wilm et al. [93] demonstrated the first idea of 1D and 2D periodicities, while taking into account that the piezoelectric was the initial effect. Analytical and numerical techniques have been used to study similar ternary 2D systems [94–99]. The research on the 2D structure discussed above concentrated on the usage of elastic waves in a solid matrix. Laude et al. [100] created a plane-wave expansion method to determine the intricate band structure of 2D infinite cylindrical inclusions in a hosting material in order to acquire a deeper understanding of the generated Bragg scattering bandgaps. By analyzing the complex value of the wave number in the frequency regions of the FBGs for each elastic polarization, they were able to properly define the dispersion of evanescent waves inside the 2D structure. Veres et al. [101] have studied the dispersion of evanescent waves in 2D phononic crystals with a square lattice of holes of various shapes.

In terms of surface waves, Meseguer et al. [102] used experimental evidence to show for the first time that the FBG exists when Rayleigh waves are considered in hexagonal and honeycomb lattices. Tanaka and Tamura [103] used a triangular lattice of cylinders implanted in a backdrop media to theoretically examine the FBG opening for Rayleigh waves in the same year. A growing interest in microelectromechanical systems was prompted by these works. In a different investigation, Liu et al. [104] created and manufactured a square lattice phononic crystal for Love waves. Meanwhile, there was a larger-scale consideration of periodic structures for seismic wave shielding. Brule et al. [105] established the first large-scale experiment in the existence of partial FBGs for seismic waves in the dispersion of Rayleigh waves induced by holes excavated in the ground.

The purpose of the NMs was to offer potential techniques for the absorbing of seismic waves of high amplitude. A few earlier research works looked into how the cross-section modeling technique was utilized to model the NMs; it was frequently simplified into a two-dimensional model. To greatly reduce the seismic waves, Colombi et al. [106] suggested using a forest as a lattice in which each tree acts as a resonator with flexural vibration. Other studies conducted on NMs consisting of a 2D lattice of forest resonators for Rayleigh waves [107,108] or Love waves [109] in substrate soil have been proposed for the reduction in seismic waves. As shown in Figure 2, a two-dimensional model was established by intercepting the plane in the x-z direction. In addition to this, the 2D model is based on the assumption of a plane strain model, which has only two degrees of freedom.

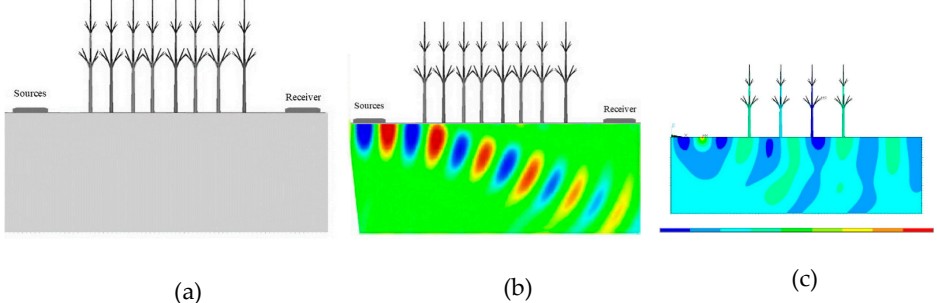

(a) (b) (c)

**Figure 2.** Two dimension of periodic trees: (**a**) NMs model; (**b**) 8 rows of NMs converting surface waves to bulk waves; and (**c**) 4 rows of NMs interacting with Rayleigh waves.

### 2.1.2. Three-Dimensional Metamaterial

In 2D representations of periodic structures, the relevant data about the entire third route are lost because the vibrations can occur in a two-axis plane only. While some data are lost when using the 2D model, these data are usually necessary for estimating the bandgap range [110]. It took until 2015 for the topic of FBGs with 3D models to gain attention, due to the tremendous advancement and wide use of additive manufacturing technology [111–117]. However, Sigalas et al. [118] finished the first theoretical investigation of band dispersion for elastic waves in a three-dimensional lattice. Liu et al. [119] completed the first experimental construction and characterization of a three-dimensional lattice using solid tungsten spheres; they reported a relative bandwidth of the FBGs of more than 50 percent, with a maximum of 75 percent. A three-dimensional lattice's FBG width was first maximized through engineering work in particular cases [105,120]. Suzuki and Yu. [121] used a computational model based on the plane-wave expansion method to determine the intricate band structure of 3D periodic structures made up of spherical tungsten scatterers embedded in a polyethylene matrix. However, this traditional configuration of spheres in a hosting medium has lost some of its appeal because embedding heavy spheres in SMs is labor-intensive, time-consuming to create, and has limited material options available, and the structure itself is heavy for use in real-world applications. The propagation of seismic waves from SMs to NMs is essentially a three-dimensional elastic wave problem. For example, Cui et al. [122] established the wave velocity in both the lateral and the longitudinal scatterer directions and the density ratio and distribution arrays, in addition to the portion of the lattice. All of those factors influenced perfectly the reported bandgap's breadth and position from 2D to 3D models of periodic structures [120]. Furthermore, higher differences in these factors will result in a wider and higher starting point for the bandgaps. In addition, the rotation of the scatterer's unit cell is one of the important factors. Du et al. looked at the differences in the out-of-plane wave FBGs caused by different scatterer rotation angles. They show that as the rotation angle is increased, the out-of-surface wave bandgap narrows first, then widens. Further modification of the bandgap according to common scatterer shapes was demonstrated as a result of FBG characteristics. Rotating scatterers are a viable option. It has been shown that the scatterers work best when they are moved around inside the lattice and also when they are rotated.

The seismic waves are a problem within 3D phases. So, the 2D model is turned into a 3D solution for damping the seismic wave which appeared clearly when calculating the FBGs. This shows how seismic waves move in an objective way. Some studies look at the SMs and how they control the bandgap at low frequencies (less than 150 Hz), which is important for controlling traffic vibrations [123–127]. Therefore, the forest is further considered as a natural large-scale phononic crystal, and a three-dimensional model was established to calculate the dispersion curve and simulate the surface Rayleigh waves [128] and objectively reflect the propagation of the Rayleigh waves. Additionally, the dynamic response of the 3D analysis of the SMs in soil was offered as an alternative to the 2D lattice analysis for the reduction in seismic waves [129,130]. J. Huang et al. [41] studied the bandgap engineering of periodic NMs at a low frequency range below 100 Hz (traffic vibration frequency range). The 3D model is used to figure out the attenuation and reduction in the seismic waves in the field of three axes (x, y, and z); so, the information about the out-of-plane wave (the third axis) is presented as shown in Figure 3.

### 2.2. Bandgap Engineering Theory

The bandgap is an interval that exists between two dispersion bands of seismic waves. The bandgaps regulated in the SMs are generally generated by two assumptions. The first is the Bragg scattering bandgap in the ultrasonic frequency ranges, where the bandgaps usually occur at wavelength (λ) of the seismic wave and are about the same as the structure order. Conversely, the second is indeed the local resonant frequency bandgap, which appears in the immediate area of the resonator's accepted inaudible frequency range, where it occurs at a wavelength much larger than the structure order. This section will

introduce different mechanisms in terms of the SMs and NMs, based on the bandgap generation mechanism.

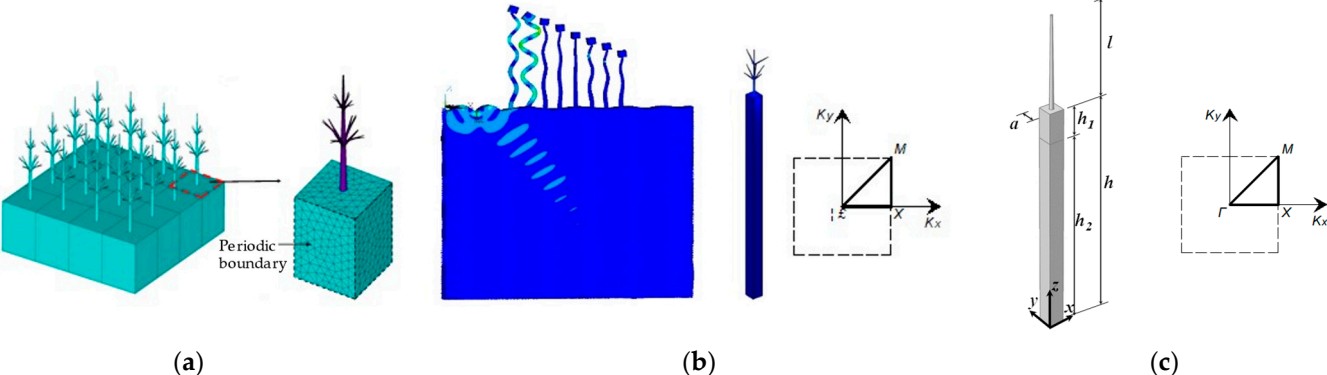

**Figure 3.** The 3D model of NMs interacting with Rayleigh waves: (**a**) The simulation model; (**b**) NMs converting surface waves to bulk waves the 3D model of; (**c**) Unit cell with viscous spring boundary.

The researchers have looked at how the scatterer's shapes affect the bandgap structure. Some of the most explored common lattice shapes are circles [63,131], squares [94,97], triangles [132,133], hexagons [134,135], obliques [136], rectangles [137,138], coated cylinders [139], and other forms [140] and are common scatterer shapes in the most recent literature. Aravantinos et al. monitored a revolutionary large-scale structure in isolated seismic vibration [141].

The bandgap achieved is substantially extended by adjusting the location of the scatterer's center and changing the scatterer's boundary measurement at all corners of the square. The binary metamaterial composite unit cell is depicted in Figure 4.

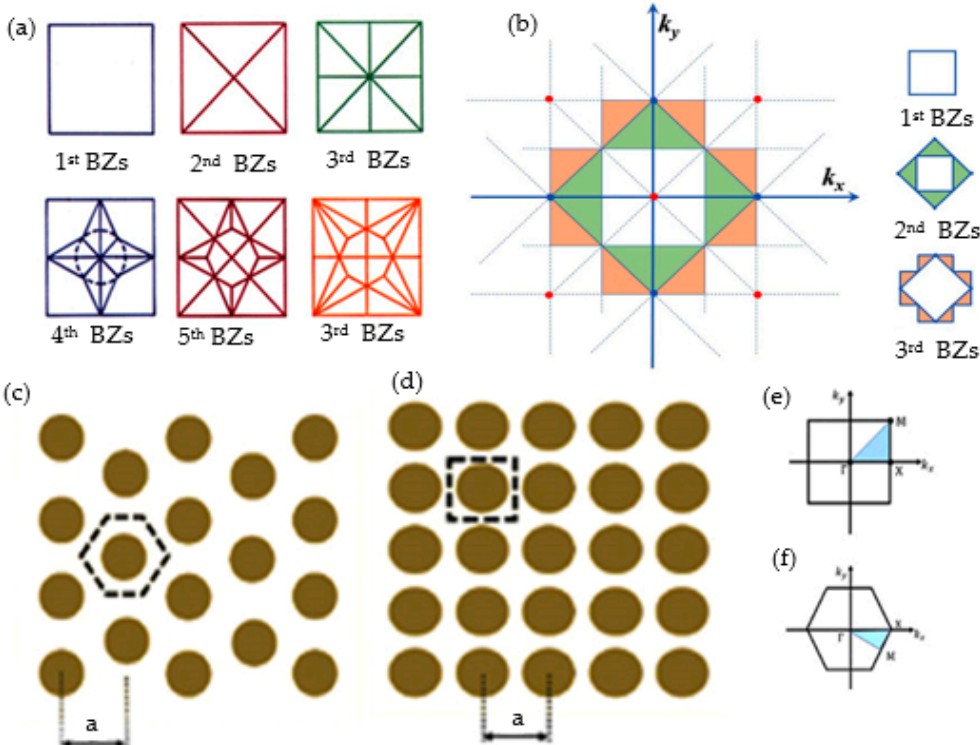

**Figure 4.** Schematic representation of lattice shapes for different vases of SMs: (**a**) square lattice; (**b**) honeycomb lattice; Irreducible Brillouin zone (IBZ) of square; (**c**) honeycomb; (**d**) lattice; First Brillouin zone of square; (**e**) honeycomb; (**f**).

The bandgap is usually measured by the distance between two portions of the dispersion curve of an elastic wave [128]. According to the significant effect of geometrical properties of the scatterers [120], the bandgap is further improved by using different shapes of the unit cell. For the natural metamaterial, the urban trees can produce bandgaps in several periodic arrangements, as shown in Figure 5.

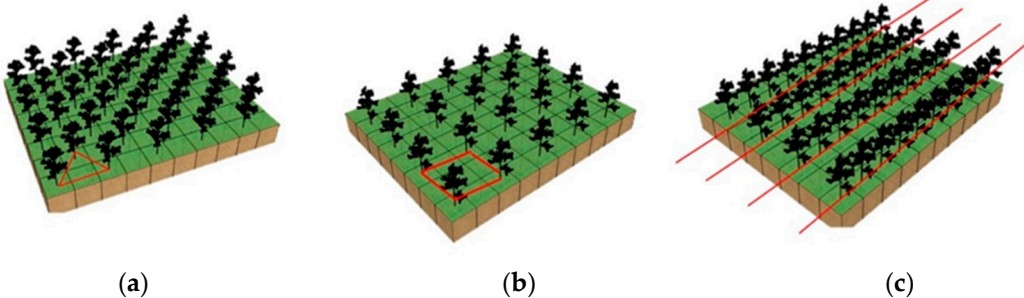

(**a**) (**b**) (**c**)

**Figure 5.** Several periodic lattices of NMs: (**a**) triangle lattice; (**b**) rectangular lattice; (**c**) square lattice.

### 2.2.1. Bandgaps of Bragg Scattering

In fact, Bragg scattering bandgaps (BSBs) were discovered a few decades ago by Brillouin in 1953. These bandgaps were created by acoustic waves dispersing in a manner similar to the way that X-rays are scattered by periodic patterns in Bragg scattering. This form of bandgap is most typically created at frequencies in the infrasonic or supersonic range, in which the wavelength is so significantly bigger than that of the lattice constant that the scatterers' material qualities will have an impact [95]. Kushwaha et al. [142] demonstrated the bandgaps of pure S-waves with an out-of-plane polarization as Bragg scattering bandgaps. Martinez-Sala et al. [143] experimentally found that the elastic wave may be observed in a sculpture bandgaps. The research into periodic structures has since become a worldwide phenomenon. Since the beginning, the primary focus of periodic structure research has been on the process that generates the FBG. The early research relied heavily on the Bragg scattering mechanism, which is referred to as the Bragg bandgap. However, the precondition for generating the Bragg bandgap is that the latticing size is in the same order of magnitude as the wavelength of the elastic wave. This bandgap is called the Bragg bandgap. The early phononic crystal research relied heavily on the Bragg scattering mechanism. Yariv A et al. [144] referred to it as the Bragg bandgap. However, the precondition for generating the Bragg bandgap is that the latticing size is in the same order of magnitude as the elastic wave. Because none of the phononic crystal's wave modes is responsive to the phononic crystal's reflection, refraction, or transmission, this frequency range is blocked from generating elastic waves, which results in a vibration attenuation domain [145]. The Bragg scattering of phononic crystals, on the other hand, requires a higher lattice size to achieve a low-frequency bandgap, which limits their practical engineering applications [146]. As the elastic wave propagates through the structure, the scatterer's self-resonance characteristics interact with the elastic wave in the matrix, resulting in the formation of a local resonance bandgap [147]. The Bragg scattering bandgap type, using a smaller lattice size, breaks, allowing the phononic crystals to be used in a wider range of applications in practical engineering in a way better than previously thought possible. Experimental and theoretical studies have demonstrated the use of three-dimensional lattices. Because none of the seismic wave modes are responsive to the periodic structure response frequency, this frequency range is blocked from generating FBGs, which results in a vibration attenuation domain. Bragg scattering, on the other hand, requires a higher lattice size to achieve wide FBGs in low frequency ranges, which limits their practical engineering applications.

### 2.2.2. Bandgap of Local Resonance

The local resonance bandgap (LRB) mechanism is another phase of the periodic structure bandgaps. Most of the previous studies have clearly produced Bragg scattering

bandgaps (BSBs); they differed in that the created FBGs are invariably in the frequency regions of the infrasonic range.

The LRB was proposed in 2000 for the first time [119]. The generated FBGs based on the LRB mainly depend on the overlapping of the resonance characteristics for the scatterers with the elastic wave medium; so, the effect of propagation for seismic waves in the structure decreases gradually. The theoretical and experimental results show that the vibration frequency attenuation range of this type of FBG, when the wavelength is bigger than the lattice constant, is essentially different from that of the Bragg scattering bandgap. The bandgap created by the LRB width is adjustable and is formed by wave scattering to create a higher frequency bandgap. When these waves both have a similar wavelength to that of the lattice constant, they have the same order. They would interact with each other and create a bandgap if certain conditions were met. It typically shows up at a frequency that corresponds to a discretized eigenmode [148].

Different types of resonators have been developed for the LRB. For example, Liu Zhengyou et al. [119] prepared an LRB from a 3D three-component resonator, which is composed of shot put balls wrapped periodically by a soft material in an epoxy resin matrix material. Hsu et al. [149] created a small, thick slab with an array of treaded resonators. Larabi et al. [150] presented a multilayer cylindrical construction. They disjointed the initial gaps into small shells that jointly generate wide bandgaps through introducing the inclusions for all of the shells. In the initial effort, Rupp et al. [151] employed topology optimization to modify the surface of a silicon substrate for Rayleigh wave filtering across a wide frequency range, resulting in FBG enlargement for the seismic waves. Colombi et al. [152] proposed spatially graded subwavelength resonators placed on the soil's surface to create a phenomenon known as a "seismic rainbow" at the geophysical scale. The resonances of the stick resonators, whose heights are gradually altered to create large LR BGs and Rayleigh waves are coupled to produce the rainbow effect. Rerouting the mechanical energy into the earth's depths to protect the structures on the surface causes the Rayleigh waves to convert into bulk waves. The rainbow effect may be created in the real world using resonators in a grove of uniformly spaced trees to provide broadband seismic shielding [109]. On the other hand, it has been established that there is a low-frequency bandgap within the noticeable range, the urban forest resonance-derived wide LRB, interacting with a lower frequency of plane waves [106]. Due to presence of the LRB, the metamaterials offer a high potential response for vibration reduction, especially at low frequencies. However, the widths of the LRB are small, suggesting practical applications for the metamaterials. Elastic metamaterials are a form of periodic structures. However, current research has discovered metamaterials with periodic units, "tree-soil", that may modify the band structures and can improve vibration attenuation [153]. D.K. Guo et al. [154] provided guidelines for the rational design of shear-wave seismic metamaterials by studying the attenuation of horizontal wave energy by the interaction of seismic resonators on layered media soils with the energy of horizontal shear waves in a half area. They explored the potential for interference-coupled resonators that could mask much of the energy of seismic waves to protect against earthquakes. Leu et al. [128] demonstrated that the associated variability of the 3D unit cell with phononic crystal lattices might lead to widening bandgaps for metamaterials. Lim et al. [108] simulated urban trees as a seismic metamaterial with branches in a variety of spatial configurations and discovered that these structures may generate wider bandgaps than those without branches. As a consequence, the BSB is often larger compared to that generated by the LRB. Moreover, the periodic structure used for both ideally permits the modulation of the FBGs at both high and low frequency bands. In other words the low-frequency vibration can be attenuated even if the lattice is small, which disrupts the limitation of the BSB on the lattice size and greatly broadens the application of the periodic structure in practical engineering. In practical situations, combining the two theorems to create an FBG structure really enables the possibility to modify the FBG both in the audio (low) and the ultrasonic (high) frequency ranges. Moreover, those structures could be used in situations where specific vibration control is required, such as crowded

cities, ground movements, roads, transportation lines, tunnels, noise, seismic waves, and construction activities.

### 2.2.3. Combination of Bragg and Local Bandgaps

It is possible to intentionally achieve the coupling between LRB and BSB and create an ultra-wide BSB coupling FBG [155,156]. This phenomenon causes concern to researchers and engineers by overlapping the BSB with the local resonance models of the SMMs rather than the modulation. The FBG's range, breadth, position on the dispersion curves, and capacity to inhibit wave propagation may all be modified by manipulating the characteristics of the periodic structures. The rising trend in practical uses based on shielding and filtering the undesired vibration in the high frequency or extremely loud frequency ranges led to the creation of a resonator for the elastic BG engineering. Despite the fact that subwavelength structures may be reached via LRBs, the difficulty of extending them must be addressed. This triggered a race to create metamaterials that could prevent the waveform across the largest possible frequency range.

A lot of research has also been conducted on how to enhance the size of FBGs by design engineering, with the early studies focusing on the lattice architectures of the unit voids in a solid material. Hussein et al. [157] created a multi-objective genetic algorithm to optimize the design of a 1D binary periodic structure for the greatest wave attenuation via broadband FBGs. Diaz et al. [158] investigated structural design networks of linked masses, where the design variable is the addition of a unit mass. They sought for the finest potential mass placement choices in order to develop the FBG, find its core frequency, and govern its width. Genetic algorithms were used to generate 2D periodic structures made of voids in silicon, and they provided ideal structural unit cells with intricate designs such as out-of-plane elastic wave FBGs and a mix of both. They were able to attain an FBG growth of 122.7 percent for out-of-plane waves [159]. For a shorter period of time, the aim has been infer a new feature that would combine LRB and BSB to be utilized in reducing undesirable vibrations and protecting vital structures from the hazards of such vibrations. The studies of MMs provide more ideas for the design the bandgap of the periodic structure; the traditional engineering structure is designed as a special artificial periodic structure based on an environment friendly work in which forest trees are the natural material, and the low frequency range is extended to 60 Hz. So, the wave propagation in the structure is suppressed by using the characteristics of the bandgap to succeed as the controlling purpose of unwanted ground vibration. By swapping waves through the first irreducible Brillouin zone (IBZ), a band structure is achieved where both the high and low FBGs are contained.

### 2.3. Propagation Mechanism

Wave propagation in periodic structures has been the subject of much study since the 1950s and 1960s, as the researchers hoped to harness the properties of the periodic structures' wave propagation to reduce vibration and noise [160]. Elastic waves are the primary mode of transmission for the majority of vibration and noise in everyday life, and their essence can be traced back to the interaction between elastic waves in a structure (material) and the surrounding media (such as air, soil, water) [161]. Controlling vibration and noise requires an in-depth understanding of the mechanisms and properties of the elastic waves in structural materials [162]. Research into phononic crystals has since become a worldwide phenomenon. Elastic waves cannot propagate in phononic crystals at certain frequency ranges because of bandgaps, which are frequency ranges in which the propagation of phononic waves is suppressed [163]. Since the beginning, the primary focus of phononic crystal research has been on the process that generates the bandgap. Liu Zhengyou et al. [119] created a three-component local resonance type of phononic crystal using an epoxy resin matrix implanted with lead balls wrapped in soft materials on a periodic basis. The results all reveal that this type of phononic crystal and the accompanying elastic wave wavelength are significantly greater than the lattice size, which

fundamentally differs from the Bragg scattering type of bandgap. There have been a few more studies that have looked at the mechanism of how the phononic crystal bandgap is formed and have attempted to estimate frequencies of the local resonance bandgap starting and ending by simplifying the phononic crystal unit cell model [164].

Typically, a phononic bandgap formed by the Bragg scattering is a periodic lattice of light. The evanescent coupling of Mie resonances of individual scatterers made of high-index materials is another mechanism for phononic bandgap formation that can be explained using the tight-binding model commonly used to explain electronic bandgaps in semiconductors [165]. Because lattice periodicity or long-range order is not required, many amorphous semiconductors have large electronic bandgaps. CBGs can also be found in amorphous photonic structures made up of strong Mie scatterers such as dielectric rods or spheres [166]. Structural periodicity during bandgap formation has been discussed with regard to how the analyzed scatterer unit interacts with the Mie scattering characteristics and how the bandgap variation law's intrinsic influence mechanism works, as has the relationship between the two bandgap mechanisms and their differences [167]. When designing a specific artificial periodic structure, the "Bragg-local resonance" coherence can be artificially realized, resulting in an ultra-wide "Bragg-local resonance" bandgap [168]. Sigalas et al. [141] theoretically established that periodic structures had a bandgap and a band structure in the early days. Martinez-Sala et al. [143] discovered the presence of a bandgap caused by elastic waves in a notable frequency range for as long as they experimented with sculptures for the first time. Based on periodic structure theory, designing artificial phononic crystals with wide bandgaps provides new ideas for reducing vibration and noise in engineering structures. Until now, the research and related works have been carried out to develop standard resolutions, such as buried and tuned walls and open trenches, as well as new strategies for improving devices and techniques based on the same principle, such as above-surface heavy barriers, biomasses or layered seismic metamaterial, and special periodic structures [169].

## 3. Toward Aiming Planting

Worldwide interest has been paid to the field of natural metamaterials. With its new type of periodic structure comes a wide range of potential applications. The plant's resonance properties are important in terms of pollution, noise abatement, and ground vibration issues. Due to the urban forest's unique properties as phononic crystals and because it has the characteristics of seismic metamaterial, it can be used as a sound barrier [170,171] and for a variety of applications, including wave converting [109], vibration attenuation [128], and noise reduction [172,173]. The purpose of this section is to investigate plant resonance in an attenuating vibration field. A portion of the wave energy received by a leaf may be lost due to friction and thus absorbed, while some of it may be re-emitted [174]. The seismic wave acting on the leaf is small at low frequencies, and the wave re-emission is likely to be inefficient [175]. The driving part of the wave energy is greater at higher frequencies, and it accounts for a significant portion of the energy received. Despite the theoretical and experimental research, the mechanisms of vibration attenuation in various plant communities remain unknown. The following advancements have focused on expanding the bandgap, low frequency, and the attainment of improved vibration attenuation based on the periodic theory concepts. The ground effect, the thermos viscous absorption of sound in the soil and in the substrate layers of the soil at the ground's surface, the scattering from trunks, branches, and leaves, and the vibration of thin branches and leaves are among the mechanisms proposed.

### 3.1. Vibration Mitigation in Urban Environment

The vegetation has effects on the propagation of the ground vibrations. The study is of the role of plants in the soil; the stems, trunks, branches, and foliage of herbs, shrubs, and trees make up the complex medium that is the vegetation offered for elastic wave propagation. The influential elements impacting on elastic wave propagation through forests and

vegetation have also been investigated and characterized using a variety of numerical and experimental methods [176]. Previous research has suggested that vegetation plays a crucial role in sound propagation via vegetation through scattering, absorption, ground effect, and reflection [177,178]. The effect of the ground motion is strong at low frequencies. As a result of direct interference between the propagation of the waves and the resonance, the ground vibrations are mitigated [179]. Because of their tiny size in proportion to the wavelength, the scattering effect of the leaves, branches, and trunks is minimal. Furthermore, at these low frequencies, the absorption from the leaves themselves is insignificant. The massive branches and trunks both scatter sound energy at mid-frequencies. At higher frequencies, often higher than 1 kHz, scattering is still important, and the foliage slows down the waves even more through viscous friction [180]. Martens et al. [181] studied the contribution of individual leaves to the attenuation of sound through generating manageable mechanical vibration at resonance frequencies so that the sound energy was converted to heat. In another research work, a laser vibrometer was used [182], as well as an accelerometer in anechoic chambers to investigate the vibration velocity of leaves. Tang et al. [183] performed similar measurements on different leaves of six different plant species: Acalyphia, laser Vibrometer device Lon, Lonicera, and Erythrina, using a light-weight accelerometer. Embleton et al. [184] employed accelerometers to analyze branch oscillations; the deciduous trees' lower branches oscillated at 300 Hz; the findings of the investigated measurements showed that the smaller branches appearing near the top of the tree have an influence at frequencies of resonance of more than 1100 Hz. The branch length was inversely proportional to the wavelength at high frequencies. Martens et al. [185] discovered that when vibration or noise pushes leaves up to 100 dB, the leaves of the trees behave in the same way as linear systems. For two reasons, left-field sound re-emissions were found to be quite minimal in the experiments. To begin with, the vibration velocity of the leaves is less than that of the particles in the air. This indicates that only a small portion of the sound energy that reaches the leaf causes it to vibrate [186]. The energy of the sound is diffracted and reflected around the leaf in the other direction [187]. Second, a leaf's complex vibration mode leads the leaves of different parts to be out of phase, cancelling the pressure vibration caused by the wind around the leaves [188]. The reduction in the curves of the frequency-absorption to two superimposes the Gaussian curves [189]. Tang et al. [183] demonstrated that the mode of the leaf's vibration can be classified into two mode types; the leaf's length belongs to the first mode type, whereas the leaf's breadth belongs to the second type of modes. This, in turn, causes the leaf's two-dimensional surface to vibrate longitudinally and transversely. The longitudinal mode of vibration causes the lower-frequency Gaussian curve, while the transverse vibration mode causes the higher-frequency curve. Because the transverse mode seems to be more prominent, it results in greater absorption. In reverberant conditions, few studies have assessed the impacts of vibrations and thermo-viscous absorption on leaves, trunks, and branches. Burns et al. [190] assessed the impact of resonance and thermo-viscous absorption on the branches and needles of the pine tree. The branch velocity was lower than the air particle velocity. At frequencies lower than 4Hz (8 cm needles) to 49 Hz, fundamental needle resonances were detected (2.3 cm needles). In order to evaluate the coefficient of absorption, Yamada et al. [191] used four different trees set in a reverberation chamber: two conifers and two with wide leaves. The values of the absorption coefficient of the broadleaf trees were found to be higher than the absorption coefficient of the conifers. According to the study findings, leaves generate acoustic attenuation; so, it does not depend on the leaf's surface area. The absorption coefficient was also demonstrated to rise proportionally to the frequency squared. At 10 kHz, the trees' highest absorption coefficient was roughly 0.2. The perpendicular vegetation system can be planted in wooden frames that make up the vertical greenery system, as shown in Figure 6.

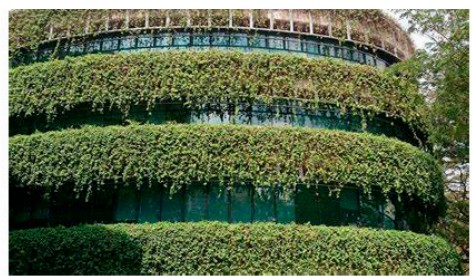
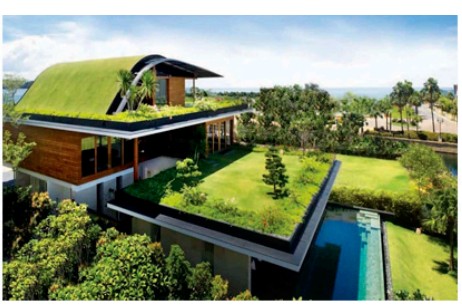

(**a**)

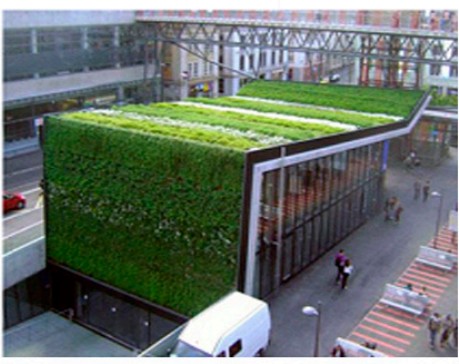
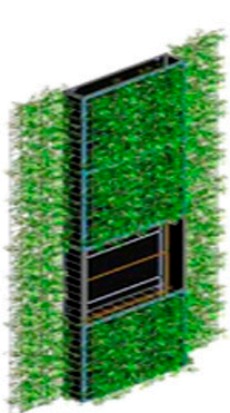

(**b**)

**Figure 6.** Green shrubs covering substrate wall applications. (**a**) Schematic representation of green buildings, with several services copyrighting the figures, https://www.hqassim.com/green-buildings/, (accessed on 22 March 2022). (**b**) Modular of the green forests covering substrate walls. Copyright for the figures, http://www.canevaflor.com, (accessed on 16 April 2022).

Padgham et al. [192] studied the frequency attenuation and reverberation decay in two woodlands. According to the results, the distance between the receiver and the source has the highest influence on the reverberation decay rate. In general, the RT grew as the source–receiver distance shrank. Compared with other frequencies, the reverberation from 1 to 3 kHz was not much different from the receiver–source distances. Reverberation is reduced when the source height is increased, while attenuation is reduced when the receiver height is increased. In a reverberation chamber, Wong et al. [193] investigated the coefficient of absorption perpendicular to the vegetation systems of a plant area. The pots were arranged inside wooden frames, and the pots had different levels of vegetation coverage (41 percent, 71 percent, and 100 percent) [194].

The increase in the coverage of vegetation improved the absorption coefficient at all frequencies from 100 Hz to 5 kHz, according to the findings [195]. There was a minimal variation in the absorption coefficient of sound between the different types of vegetation at low frequencies; yet, lower than 1 kHz, the contrast in sound absorption coefficients is generally continuous [196]. Kaye et al. [197] investigated the absorption coefficients of sand, snow, gravel, ashes, turf, and railway track ballast. The absorption coefficients rose with the increasing frequency, according to the findings [198]. The values of the absorption coefficients were high. Reethof et al. [199] used an impedance tube to study the trees' bark and assess their normal incident absorption coefficients. The absorption coefficients of most species are frequency independent. The theoretical and numerical investigation of the protective effect of forests as locally resonant metamaterials was conducted. Elastic wave propagation and unwanted vibration attenuation via dispersion relations and frequency response studies showed the unblemished importance of plantation in urban cities. NM researchers at Imperial College London have made significant contributions by replicating and transforming the wave phenomena seen in EMs into seismology and geophysics.

### 3.2. Naturally Available Metamaterial

Metamaterials (MMs) are a new type of artificially designed material that has qualities that are not found in naturally existing materials [200]. Rather than the inherent qualities of the underlying materials, these supernatural properties are obtained from their specially engineered microstructures. They are frequently made up of single- or multi-phase conventional material elements that are arranged in low-dimensional repeating patterns. Engineered periodic structures with artificially fabricated extrinsic and low-dimensional inhomogeneity can change seismic wave speed, direction, and wavelength, leading to fascinating features [201]. From an early time, much research focused on the role of metamaterials in ground vibration mechanisms. However, scientists did not begin studying MMs until recently due to the discovery of negative permittivity and permeability in some artificial materials, which have yet to be found in nature [202]. As a result of this idea, engineers have begun to develop new materials with properties that exceed the capabilities of the conventional ones. Recently, the acoustic/elastic wave (EWs) counterparts of electromagnetic metamaterials have been developed [191,203,204]. Similarly to electromagnetic wave propagation, MMs were first considered in electromagnetic wave propagation [205]. Elastic metamaterials (EMs) are the primary focus of the research into metamaterials because they make use of a resonance between their unit cells and an external electromagnetic field. EMs are also called optical metamaterials [206]. Elastic wave scattering can be controlled using either EMs or optical metamaterials, allowing them to be used as natural elastic wave media in a wide range of applications, such as vibration [113] and noise reduction for EMs, together with magnetic force absorbers [207], acoustic cloaking [208], negative refraction [209], unidirectional and subwavelength waveguides [210], backwards wave antennation [211], and permanent magnets with high performance [212]. However, some theories of EMs have been questioned by scientists. As an example, Markel et al. [213] believed that negative index materials would lead to alternative issues such as negative heat, which is not possible in physics. As EWs propagate photonic crystals as elastic waves in the same medium, the application of EMs to control elastic wave propagation comes to mind [214]. EMs are specially engineered periodic structures that have the positive properties of MMs, such as structural and periodic density variation, as well as high anisotropy, which can be obtained by designing the minimum periodic elements and the overall configuration in a reasonable manner. Each of the theoretical and experimental problems with MMs represents an additional area for the studies to clear up. The authors demonstrated that the efficiency of the vibrations in the reduction in undesired waste rises as the number of tree lines increases and that they (the trees) may be seen as a large-scale natural forest metamaterial through simulation. Colombi et al. [106]. found that forest trees on the ground act as subwavelength MMs and induce subwavelength FBGs in a wide frequency range of interest. Although the reported result was convincing that forest trees can be considered naturally available SMMs, the concept of the MMs reported conflicted with the conventional meaning, as shown in Figure 7.

They found a strong attenuation phenomenon when the wave propagated with a frequency of less than 150 Hz after passing through two detached bandgaps in soft soil as a result of the resonance of trees across the subwavelength of the incident Rayleigh wave. The use of forests as locally seismic metamaterial resonators on Rayleigh waves appears in another study. Maurel et al. [109] examined the propagation of Love waves in forest trees and reported Love-wave FBGs. The surface shear horizontal wave interaction with the translation mode of the resonators induces FBGs, which prohibit the penetration of surface Love waves. The inspected Love waves were scattering through a forest at the top of the soil substrate guiding layer. They discovered that the trees' foliage causes a fundamental change in the dispersion relation nature of the surface waves. Huang, J. et al. [128] proposed a new and different approach to analyze the use of trees to reduce ground vibrations using urban forests. Urban trees are implanted on the basis of periodic arrangement in the spatial engineering configuration; the resonant system of trees is considered as a cantilever bending the beam below the electromagnetic wave propagation at specific

frequencies. Mohammed et al. [107] studied urban trees as SMMs and the effect of branches in simulations to produce wide bandgaps of Rayleigh wave propagation. The existence of wide FBGs in seismic MMs theoretically and experimentally opens the door for researchers to conduct interesting studies in natural materials with the same purposes. For example, the Rayleigh wave and the resonant modes of resonators have produced FBGs to mitigate the propagation of surface waves in a certain frequency range at the resonator attachment point.

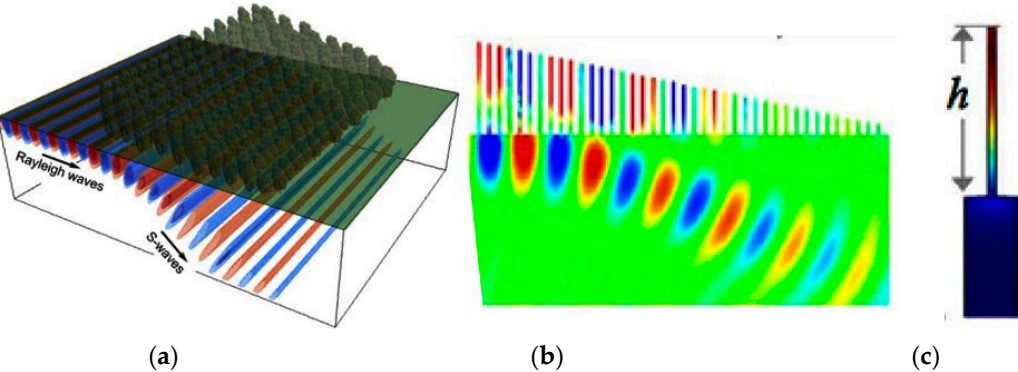

(**a**)　　　　　　　　　　　　　　　(**b**)　　　　　　　　　　　　　　　(**c**)

**Figure 7.** The conversion phenomenon of NMs as rendered from the 2D numerical simulations: (**a**) artist view; (**b**) displacement plot; (**c**) unit cell. Figure reproduced from [215].

It is actually possible to use the common tree height of green trees in urban green forests (between 8 and 15 m) as a subwavelength local resonance unit, given that the surface wave wavelength is typically between 5 and 10 m [107]. Lim et al. [107] comprehensively simulated at the geophysical scale that the Rayleigh wave propagation through soft sedimentary soil basins involves strong wave attenuation when the longitudinal resonant modes of trees are involved, and they conclude that the resonances of trees with correct alignment conditions precisely collaborate as locally available resonant SMs that are furnished with wonderful capabilities. In another study, simulations have revealed that bulky side-branched trees interact with Rayleigh waves, causing a wide local resonant bandgap to form in the earthquake frequency range of interest [108].

Because none of the seismic wave modes is responsive to the SM's reflection, refraction, or transmission, this frequency range is blocked from generating seismic waves. Martnez-Sala R et al. [216] pointed out that the regularly ordered green belts can be considered to be phononic crystals; it has not been obvious how the vibrations are reduced and what model to use. Follow-up research on green forests' vibration-reducing mechanisms has been the subject of some studies in the last few years. Green woods, according to this theory, can produce an "earthquake rainbow" effect comparable to an electromagnetic surface optical rainbow by converting incident surface Rayleigh waves into bulk elastic shear waves or reflected Rayleigh waves. Elastic waves can be reflected back or changed into body waves when they travel through a green forest with increasing or decreasing heights [217].

As a result of the 2D model's simplification of trees as large-scale phononic crystals with attenuation zones, low frequency, and wide FBGs turned into multi-row "tree walls", which ignored the absence of vibration outside the plane, trees now have a more realistic 3D appearance. Planting trees of varying heights on the ground will result in a strong invasion in a highly urbanized environment, even though the outer layer shielding SMs on the ground are simple to structure and arrange. Reduced traffic vibrations and structure protection in a lower frequency range may theoretically be achieved by a green forest in metropolitan areas [128]. Bragg scattering SMs require a higher lattice size to achieve a low-frequency bandgap, which limits their practical engineering applications. Thus, this behavior was described as a bandgap in phononic crystals to stop the propagation of Rayleigh waves at the corresponding frequencies and the interaction of trees with Rayleigh waves in soil, but they ignored the effect of the trees' branch properties due to their small weight compared to the weight of the trunk [41,153]. They developed a 3D simulation

model using forest metamaterials as the basis to investigate the protective effects of the 80 Hz low-frequency Rayleigh wave. The city green spaces are considered to be large-scale in terms of natural metamaterials, as shown in Figure 8.

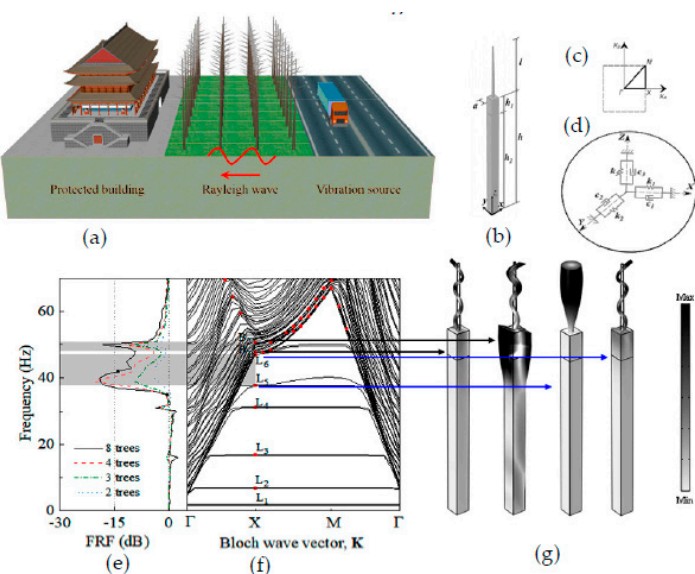

**Figure 8.** Urban forest as large-scale NMs: (**a**) Rayleigh waves and periodic arrangement of forest trees; (**b**) unit cell; (**c**) wave vector in first IBZ; (**d**) viscous spring boundary, (**e**) FRF plot, (**f**) dispersion curves with FBGs, and (**g**) vibration modes corresponding with FBGs boundaries. Figure reproduced with permission from [128].

As the elastic wave propagates through the NMs, the scatterer's self-resonance characteristics interact with the seismic wave, resulting in the formation of a local resonance bandgap. Experimentally and theoretically, the results all reveal that NMs are accompanying seismic waves when the wavelengths are significantly greater than the lattice size, which fundamentally differs from both types of bandgap. Using a smaller lattice size breaks through the bandgaps, allowing NMs to be used in a wider range of applications in practical engineering than previously thought possible.

### 4. Governing Mathematical Equations

The mathematical meaning of the research methods has developed according to the alleged wave hybridization phenomena between incident waves and the resonator in the soil. Here, the propagation of the EWs in the proposed models is considered, as in the following equation [218].

$$\frac{E}{2\,(1+v)}\nabla^2 u + \frac{E}{2\,(1+v)\,(1-2v)}\nabla(\nabla.u) = -\rho\omega^2 u \tag{1}$$

Unit cells should gratify the boundary conditions according to Bloch theory.

$$u(r+a) = e^{ik.a}u(r) \tag{2}$$

where r is the position vector connecting the matching points on the unit cell's boundary; u(r) is the vector of displacement of the nodes; a is the lattice transformation vector. The models should satisfy the periodic boundary conditions, as in Equation (1).

Then, by substituting Equation (1) in Equation (2), the eigenvalue can be written as:

$$(\Omega(K) - \omega^2 M(K)).u = 0 \tag{3}$$

where $\Omega$ is the global stiffness, and M is the mass matrix of the unit cell; these are functions of the wave vector, K.

Ground vibration attenuation occurs on the bottom and right sides of the resonators [33,55,219]. Floquet–Bloch periodicity requires that the unit cell be viewed as a periodic structure. The presence of bulk waves with mixed wave polarizations complicates surface wave studies in the finite domain. Elastic waves can only propagate if their wavelengths are greater than the depth of the subsoil. The material properties of the matrix have an important influence on the dispersion curve. The main three parameters included are the elastic modulus, Poisson's ratio, and density. Other properties can be calculated from these three parameters; meanwhile, the wave velocity of the material is equal to $v_p = \sqrt{\frac{E}{\rho}}$ , $v_s = \sqrt{\frac{\mu}{\rho}}$, where $V_p$ and $V_s$ are the wave velocity components, E is the Young's modulus, $\mu$ is the shear modulus, and $\rho$ is the density. The wave vectors are calculated from the minimum phase velocities of the different propagation directions in the matrix. The width and position of the bandgaps are affected by resonator factors such as ground qualities, height, thickness, foliage, spacing, eccentricity, lattice shape, configurations, and the distance from resonance system to the sources of vibration. The derivative of a periodic function based on Bloch's theorem with symmetry and traction-free condition has been applied to the top part of the unit cell. For a given wave vector, the existing circular frequency, $\omega$, can obtain the dispersion curves of the elastic surface waves [220]. Several mathematical schemes and techniques could be used to calculate the swapping vector of elastic surface waves propagating in a given frequency range. To avoid numerical calculations, the finite element method is used to approximate the solution within a given elastic domain as a combination of the fundamental solutions generated by a collection of virtual sources positioned outside the domain.

## 5. Parametric Study Comparation

The geometrical layout of the component materials, as well as the differences in their mechanical characteristics, serve as an evident technique for increasing the FBGs, which has a significant influence on seismic wave attenuation [221]. The studying of the geometric parameters and mechanical properties of the periodic structures took the first place in the objectives of the metamaterial and phononic crystal research in order to find the factors affecting the bandgap generation [222,223]. Many field tests were carried out in situ to identify the function of the mechanical characteristics and geometric parameters of the MMs, collectively and individually, in order to demonstrate the effect of the variables on the width and position of the created FBGs. This section presents a review of the results of the most important influencing factors, summarized for a comparison of the main findings of the previous studies that intensively studied the most important factors affecting attenuation mechanics, taking periodic piles as seismic metamaterial and resonant trees in 3D and 2D structures as natural metamaterial into account. As previously mentioned, a metamaterial-based seismic shield in periodic system barriers was described, and substantial degradation inside the bandgaps was numerically proven. In large-scale studies, periodic configurations of the SMs revealed attenuation tendencies in the Bragg scattering region for anti-earthquake applications.

The ground properties vary, and according to geological findings, the soil properties change every 30 feet [224]. Bandgaps are induced by the local resonance of the metamaterial [119]. Low-frequency bandgaps are induced in loose soil media where the Rayleigh wave velocity is relatively slow by the resonator. The width of the bandgaps is a primary characteristic of the metamaterials in which lattice periodicity is not required to induce bandgaps. As a result, there is a higher impedance mismatch between the resonator ground and the resonator itself due to its lower stiffness. Another factor affecting the strength of the wave modes' interaction between the resonator trees and the earth is the impedance mismatch [140]. Noteworthily, the considerations for the parametric study are tabulated within a comparison of the three different study insets, and conclusions can be drawn for the greater FBG regime. The bottom lip of the bandgap shrinks and the overall FBG breadth grows as the resonator height grows from 8 m to 12 m. The top portion of the first FBG gradually reduces as the tree height grows, while the bottom of the FBG rapidly

decreases. Then, the gap width increases to a maximum and then decreases. Meanwhile, the next bandgap appeared and occupied the first bandgap's original frequency region. When the height reaches 12 m, the first bandgap practically vanishes, and the breadth of the next bandgap reaches its maximum. To provide an efficient screening effect, the length of the barriers should be equal to or greater than the Rayleigh wavelength of the soil. According to Maural et al. [152], when the urban tree is taken into consideration, the cut-off frequencies are greatly reduced, which is already true for the spoof plasmons in the absence of a guiding layer. With the increase in height, the narrowing of the FBGs is caused by the appearance of additional modes and will eventually close [225]. Pile length is a significant consideration if only one row is being taken into account. Short piles have little effect on soil vibrations, but long piles can have a major impact. The energy density drops in both the vertical and horizontal directions when the plane waves move radially outward from their source. Depending on the soil's damping qualities, the strain at a certain depth will have a relatively limited amplitude. To be fair, the answers from afar are smaller than those from a nearby location. Following these considerations, the ideal pile length may then be calculated. Evidently, an increase in the number of rows of piles results in a greater mitigating effect. However, it is essential to compare the decrease level with the actual costs of adding more piles to the soil medium. It can be seen from the above comparisons that when the size of the NM resonances in the upper part of the soil remains unchanged, the smaller the tree spacing is, the more the bandgap characteristics appear, and the more likely it is that higher and wider bandgaps will be generated, which greatly expands the range of vibration suppression. This is an instruction to properly plant densely when planting urban green forests. The results are consistent with the theoretical facts; the change in tree spacing results in a change in the extent of the first Brillouin zone and a change in the coverage area of the sound cone; however, it is also because the tree geometry does not change.

Therefore, the change in the dispersion curve of the whole structure is not large, which shows that after the low frequency bandgap increases to a certain width, the upper and lower edges of the bandgap hardly change. The study is of the effect of the mechanical properties and geometric parameters from the previous research that dealt with the most important factors affecting the characteristic of FBGs. The purpose of this work is to open the door for future research to build the best model of natural metamaterials easily with the external features of afforestation areas and urban cities, with slight improvements for the controlling of seismic waves and the mitigation of the ground vibrations, as shown in Table 1.

**Table 1.** The main factors on the FBG width and the locations of various metamaterials.

| Source of Study | Factor | Implications and Drawbacks |
|---|---|---|
| *Ecological Engineering* (Van Renterghem, 2014) [226] | 1. Ground Effect<br>2. Source–receiver distance<br>3. Trunks<br>4. Crown scattering | • The ground effect largely affects lower frequencies at common receiver heights.<br>• The best way to lessen road noise is to put tree/ vegetation belts near to the source.<br>• Tree trunks will cause sound to be reflected, diffracted, shielded, and dispersed. Trunks offer direct shielding of higher sound frequencies, but noise reduction is not achievable at very low frequencies by tree trunks.<br>• The effect of crown scattering is mostly negative, and it is most pronounced when the sound frequency spectrum is dominated by greater vehicle speeds. |

**Table 1.** *Cont.*

| Source of Study | Factor | Implications and Drawbacks |
|---|---|---|
| *International Soil and Water Conservation* [128] | 1. Lattice spacing<br>2. Height<br>3. Number of periodic rows<br>4. Elastic modulus of soil<br>5. Poisson's ratio of soil<br>6. Bulk density of soil<br>7. Radius of resonator tree. | • When the spacing continues to decrease, the original low frequency band keeps increasing. When it increases to a certain width, a new high frequency band begins to appear.<br>• FBG does not begin to appear until it reaches a wavelength equivalent to the wavelength of the propagating waves. As the tree height increased, so did the overall attenuation domain.<br>• The vibration attenuation becomes more noticeable as the row number increases, especially for four or eight rows of trees.<br>• The elastic modulus of soil plays a more important role in the dispersion curve structure; it determines the overall coverage area of the sound cone and is a subsiding issue. A higher soil elastic modulus has a wider frequency vibration reduction capability, thereby obtaining a better vibration reduction effect.<br>• Soil density has a small variation effect; it does not have one as strong as that of the elastic modulus.<br>• Poisson's ratio has a small variation effect; it does not have as strong as that of the elastic modulus.<br>• As the periodic structure radius/thickness increases, each bandgap goes through a process of appearing, growing, and finally disappearing. This is complemented by the bandgap. |
| *Repository* [227] | 1. Distance between the pile and the excitation source.<br>2. Length of the pile<br>3. Number of rows<br>4. Net distance between piles | • Periodic structures positioned closer to sources of excitation are more effective in reducing ground vibrations, especially for soft soils with slow shear wave velocity.<br>• The length of the pile is insufficient to attenuate the incident waves. The attenuation response of the 15 m case is similar to that of the 20 m, 25 m, and 30 m cases. In addition, the study shows that the 30 m pile has double the volume of the 15 m pile; its overall reduction efficiency is just 1,2% greater. In the frequency domain, all examples analyzed exhibit decrease. The effect of the pile length on the resonant response begins at a length of 6 m, and an increase in length of more than 20 m has a negligible effect on the resonant response.<br>• An increased mitigating effect can be seen when the number of pile rows increases.<br>• The reduction efficiency increases with a decrease in the net distance between piles. Notable is the fact that multiple rows of piles do not ensure a drop in reaction proportional to pile volume. |

**Table 1.** *Cont.*

| Source of Study | Factor | Implications and Drawbacks |
|---|---|---|
| *AIP ADVANCES 7* [228] | 1. Ratio of hollow thickness to the lattice (b/a). 2. The influence of the steel pile thickness (b−c)/a. 3. Unit cell height(h/a). 4. Young's modulus and mass density of soil. | • They studied the performance of the metamaterial comprised of hollow square piles, the influence of the ratio b/a on the width of FBGs of Lamb waves; they found that as b/a increases, the first CBG size evidently increases, while the second CBG size increases initially, but remains stable beyond b/a = 0.7. <br> • The first FBG widens slowly as the thickness increases, whereas the size of the second FBG decreases initially, but remains almost unaltered beyond (b−c)/a = 0.25, and the optimal parameter is around 0.45. <br> • The height has an important effect in high-frequency bandgaps; their calculations demonstrate that the unit cell height influences suspended frequencies that are lower than the other factors evaluated. <br> • The FBG evolution trend reveals that the soil's Young's modulus and mass density have a major influence on the location and width of FBGs. These findings also suggest that it is critical to examine the mechanical properties of soils at various construction sites in order to attain the necessary FBGs. |
| *International Journal of Structural Stability and Dynamics* [107]. | 1. Number of resonators 2. Wave velocity in substrate soil 3. Wave velocity inside the resonator 4. Height, width, and spacing | • The performance of vibration attenuation depends on the number of resonators N embedded in the ground. The amount of attenuation increases as N increases. <br> • The velocity of waves propagating in the medium has a large influence on the coupling of surface waves with resonant modes of resonators. It influences the intensity of wave interaction together with the position of the eigenmodes that define the lower and upper bandgap edges. <br> • The higher the Rayleigh wave's velocity, the higher the frequency at which the resonant modes resonate. As longitudinal modes are responsible for creating bandgaps, firmer soil transfers the longitudinal modes to higher frequencies while traveling at a higher speed. <br> • Water content, species, and grain distribution all have an impact on wave velocity on the resonators. The variations in wave speed are caused by the mechanical characteristics of the resonator. The generation of bandgaps is affected by these fluctuations in the resonator's wave speed, which affects wave hybridization. Another key factor in determining the location and width of bandgaps in resonators is the longitudinal and shear wave velocity. <br> • Eigenmodes decrease as bandgaps shift to lower frequencies with increasing resonator tree height. The bandgap widens as the resonator's height rises. In addition, the bandgap leaps from the lower eigenmodes to the higher resonant modes as the frequency increases. Attenuating low-frequency Rayleigh waves requires taller trees. As the mass of the resonator increases due to the increased moment of inertia, a thicker trunk has a greater moment of inertia. This results in a system with higher stiffness and larger bandgaps. To put it another way, as a tree grows taller and wider, its resonant modes tend to become more low-frequency. |

## 6. Conclusions

The development of seismic metamaterials to connect media in terms of controlling ground vibration with natural metamaterial confirms the study of the tree resonances in that it results in forests acting as locally resonant metamaterials for elastic waves at the geophysical nano-scale. Urban green forests are a very important technology in city planning, especially when green forests are planned in a periodic manner. On the basis of the research theories and methods of computation in materials and other related fields, the periodic planning of green forest farms by creating dynamic models for the analysis of seismic resonant materials within the framework of the theory of periodic structures has three aspects: theoretical computation, numerical simulation, and field tests of vibration measurement. Vibration reduction mechanisms have been comprehensively reviewed and verified, and a set of methods for calculating bandgaps in periodic structures from seismic metamaterials to natural metamaterials are reviewed; their benefits in elastic wave isolation and control have been realized. This review demonstrates that the elastic wave, propagating in soft sedimentary soil at low frequency, experiences attenuation techniques when interacting with a natural metamaterial over wide bandgaps. The interest in the width of bandgaps and location has been presented in the outcome of this study. The seismic shielding conundrum by natural metamaterials presents great challenges that must be determined economically, practically, and thoroughly through an expansion of the practical application and the in-depth research of natural metamaterials. The main points of this paper are summarized as follows:

- Through comprehensive observation of vibration patterns there are Bragg scattering-type bandgaps and local resonance bandgaps, due to the periodic arrangement of the MMs, causing the elastic wave to be reflected, refracted, and deflected at lower frequencies so that propagation is suppressed at the specified frequency.
- In this paper, the urban green forest is considered as a periodic structure (super seismic resonators) through the method of dynamic computation of the periodic attenuation field of urban green forests based on the periodic theory, and the paper verifies the effectiveness of the attenuation by external vibration tests, providing ideal theoretical support for urban vegetation formation with vibration and noise reduction functions.
- The mechanical properties of soils and resonators play a major role in the scattering of the directed waves, and the ideal controlling properties are the elastic modulus, density, and Poisson's ratio.
- The geometric properties of the resonator's height, diameter, distribution lattice constant, number of trees, distance from the wave source, and the height of the studied soil model are the most influential properties in the width and position of bandgaps and in determining the starting point for those FBGs.

Based on the major findings of this review, several perspectives are recommended:

- Adopting and enhancing urban design approaches for cities and infrastructure to safeguard them from ground vibrations, especially in tiny spaces surrounding facilities.
- Encouraging the investigation of alternative plant types and the application of adequate soil to provide acceptable vibration damping in the fewest possible rows of super-resonant materials.
- Researchers in the fields of the environment and forests should perform experiments and ecological insights on the development of natural resonant super-materials with significant seismic damping to protect city buildings.
- Consideration of how natural resonant metamaterials can reduce noise, vibration, and other pollution sources feature in producing wide bandgaps for both high and low frequency ranges.
- Natural metamaterials can minimize the cost of synthetic ones, last longer, and attenuate vibrations and surface and particle waves better.
- With natural growth and climate change, industrial activities, engines, and generators in all sectors, including cooling in the summer and heating in the winter, may generate

additional noise and disturbance and must be reduced; natural metamaterial may help limit this spread.

- The appropriate technique is to arrange forest green belts periodically around protected constructures, along transportation lines, and in crowded areas of cities.

**Author Contributions:** Conceptualization, A.-S.Q., J.H., M.A., D.N.Q., A.-D.W. and R.F.; methodology, A.-S.Q., J.H., M.A., D.N.Q., A.-D.W. and R.F.; software, A.-S.Q., J.H., M.A., D.N.Q., A.-D.W. and R.F.; validation, A.-S.Q., J.H., M.A., D.N.Q., A.-D.W. and R.F.; formal analysis, A.-S.Q., J.H., M.A., D.N.Q., A.-D.W. and R.F.; investigation, A.-S.Q., J.H., M.A., D.N.Q., A.-D.W. and R.F.; resources, A.-S.Q., J.H., M.A., D.N.Q., A.-D.W. and R.F.; data curation, A.-S.Q., J.H., M.A., D.N.Q., A.-D.W. and R.F.; writing—original draft preparation, A.-S.Q., J.H., M.A., D.N.Q., A.-D.W. and R.F.; writing—review and editing, A.-S.Q., J.H., M.A., D.N.Q., A.-D.W. and R.F.; supervision, A.-S.Q., J.H., M.A., D.N.Q., A.-D.W. and R.F.; project administration, A.-S.Q., J.H., M.A., D.N.Q., A.-D.W. and R.F.; funding acquisition, A.-S.Q., J.H., M.A., D.N.Q., A.-D.W. and R.F. All authors have read and agreed to the published version of the manuscript.

**Funding:** The authors gratefully acknowledge that: "This project was supported by the Deanship of Scientific Research at Prince Sattam bin Abdulaziz University under the research project No: 2022/01/19950". This work was supported by the National Natural Science Foundation of China (Grant Nos. 32071841 and 31700637).

**Institutional Review Board Statement:** Not applicable.

**Informed Consent Statement:** Not applicable.

**Data Availability Statement:** Not applicable.

**Acknowledgments:** I gratefully acknowledge the Beijing Municipal Education Commission for their financial support through the Innovative Transdisciplinary Program "Ecological Restoration Engineering" and Beijing Government of Forbidden City Scholarship.

**Conflicts of Interest:** The authors declare no conflict of interest.

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
