# Peer review of "Seismic Composite Metamaterial: A Review"

_jcs, doi:10.3390/jcs6110348_

Round 1
Reviewer 1 Report
This review article presents a detailed exposition of the recent research progress of seismic metamaterials, including 2D, 3D and the main mechanisms of theoretical backgrounds of energy attenuation. In addition, natural metamaterials and the study of urban environment are surveyed. The authors could consider to cite a recent paper on seismic metamaterial on Love wave, D.K. Guo and T. Chen*, Seismic metamaterials for energy attenuation of shear horizontal waves in transversely isotropic media. Materials Today Communications 28, 102526 2021. http://doi.org/10.1016/j.mtcomm.2021.102526. In page 19, the longitudinal wave velocity Vp is not exactly correct, see for example the text book of Graff or Achenbach (1973). There is also a typo of Poisson's ratio in the Abstract.
Reviewer 2 Report
1. Addition of mathematical models to explain each mechanism would help offer more context and provide stronger scientific foundation.
2. Images and graphs do not seem to be very clear or legible. Perhaps submitting a higher resolution will help with readability.
Round 2
Reviewer 1 Report
The revision is satisfactory. I have no further comments. The manuscript can be published as is.